# Unsupervised Morphological Segmentation for Hybrid Tokenization in Uzbek Language

## Abstract

The complex agglutinative structure of the Uzbek language makes it difficult for conventional tokenization approaches to tokenize words correctly due to the high "Out-Of-Vocabulary" (OOV) rate with conventional word-level tokenization and the loss of semantic importance with character-level tokenization. This study presents a new method for creating hybrid tokenization algorithms for Uzbek by using unsupervised morphological segmentation to create a hybrid tokenization strategy. By applying the Morfessor algorithm, we trained a morphological segmenter using a large-scale corpus of Uzbek. Using the Minimum Description Length (MDL) principle, the Morfessor algorithm was able to identify the linguistic boundaries of a word's root and its various functional affixes without requiring annotated data for training. Our preliminary experiments show that the Morfessor algorithm can successfully segment complex Uzbek words into their constitutive morphemes (for example, 'maktablarimizdagilar' (English: 'those who are in our schools') can be separated into 'maktab' (school) + 'lar' (plural) + 'imiz' (our) + 'dagi' (in) + 'lar' (plural)) thereby reducing the level of vocabulary sparsity. The research conducted here represents the first step towards creating hybrid Uzbek tokenization algorithms to be used in NLP (i.e. LLM) and NMT models. This hybrid tokenization strategy will provide a scalable and linguistically valid alternative to subword segmentation methods (i.e. BPE) used in low-resourced language settings and will improve the morphological integrity of the tokenized words.

## 1  Introduction

Computational linguistics combines statistical, machine learning, and deep learning models with human natural language modeling using rule-based techniques. An essential part of natural language processing (NLP) is morphological analysis, which forms a core part of linguistic study by examining the structure of words. This analysis offers insights for every word as part of overall NLP tasks, producing properties such as the "stem", "root", and the morphological role of its "suffixes" [1].

The majority of human language technology is challenged by the fact that some languages primarily express grammatical meaning through the use of suffixes and prefixes. In these morphologically rich languages, such as Uzbek, morphemes—the meaningful components of words—are widely employed in word construction, resulting in a rich morphology with numerous exceptions. According to its word formation structure, an Uzbek word includes prefixes, suffixes, and base words. Normally, these words consist of a root form and affixes attached to the root [2]. Morphology studies how these words are built and changed to mean different things, generally categorizing into inflectional and derivational morphology, both of which are significant in different facets of NLP. To express grammatical nuances, an inflectional word ending—one or more suffixes—is concatenated to the end of a word, modifying only the lexical and grammatical meanings without affecting core meanings. Because the Uzbek language is highly inflected, the large number of suffixes resulting from homonymy causes many problems. As you would expect, when the number of suffix combinations increases, the number of probable analyses of a single word increases significantly [3].

Historically, the goal of morphological analyzers has been to identify the root and suffixes of a word. One methodology involves developing an inflectional ending dataset tagged with morphological feature tags to implement neural network models. Another goal is to develop a morphological analysis model to extract information like stem, lemma, and morphological features using a complete set of

endings (CSE), enriched by morphological rules. A common method is the stemming approach, which finds the word's root form by removing affixes. For instance, the MorphUz analyzer relies on an Uzbek stemmer algorithm that specifically focuses on identifying and separating suffixes from the stem. MorphUz utilizes a lexicon containing around 80,000 stem morphemes categorized by word groups. It achieves analysis in two steps: first, it compares each input text with lexicon stem words letter by letter from left to right; if the input word is not found in the lexicon, it applies a second step to find suffixes from a predefined list in a database [4].

While rule-based stemming and lexicon-dependent methods provide valuable morphological insights, they face critical bottlenecks when scaling to modern Natural Language Processing (NLP) architectures, particularly Large Language Models (LLMs). Conventional word-level tokenizers suffer from an exponentially growing vocabulary size and severe "Out-Of-Vocabulary" (OOV) rates in such highly agglutinative contexts. Conversely, purely statistical subword tokenizers (like Byte-Pair Encoding) often split words strictly based on frequency, destroying the very morphological integrity that systems like MorphUz attempt to parse. To address this reliance on exhaustive, manually curated lexicons and manual tagging, we propose an unsupervised, data-driven approach to construct a hybrid tokenization strategy via the Morfessor algorithm, tailored specifically to seamlessly identify the morphological boundaries of the Uzbek language.

## 2 RELATED WORK

Morphological analysis is a core part of Natural Language Processing (NLP). While a number of scientific studies have investigated morphological segmentation methods to reduce corpus size in inflectional languages, research specific to Uzbek morphology remains predominantly theoretical, with only a few practical morphological tools having been developed. However, it is noteworthy that recent NLP works have made efforts to reduce Uzbek's label as a low-resource language by introducing fundamental resources such as stop words datasets and parallel text datasets for machine translation model creation [5-6].

To process the complex morphology of the language, several models using finite state machines (FSM) have been proposed for morphological analysis in Uzbek. Researchers have developed morphotactic rules and rule-based morphological analysis models. Some approaches employ automata to extract morphological information based on established grammatical rules, utilizing a database of word forms rather than relying solely on grammatical principles. Additionally, FSM models have been used to investigate stemming with an emphasis on retrieving pertinent morphological information. Despite these efforts, the complex structure of Uzbek words presents numerous exceptional cases. Issues such as homonymy, synonymy of affixes, and vowel harmony can lead to errors in morphological segmentation and ultimately decrease the accuracy of FSM-based analysis models [7-8].

Among the few morphological analyzer tools developed for the Uzbek language, UzMor, Uz-Kaz-NLP, and the UM analyzer have demonstrated very good results, though there remains room for improvement. UzMor, for instance, is designed to find the root form of a word and can be applied to search engines to improve the retrieval of documents containing Uzbek text. It relies on a dictionary base of 30,000 roots as initial data, which allows for the generation of more than 1,500,000 word forms. The algorithm operates through a strict sequence of steps: first, the word is searched in the dictionary of initial forms. If not found, the program attempts to identify attached suffixes by reading character by character in reverse order (starting from the end of the word), removing the longest matching suffixes until up to three are truncated. Based on the stemming results from the previous step, a search for the stem is performed in the stem lexicon; if a match is found, a dictionary consisting of the word stem and truncated suffixes is returned to the user, otherwise, the process loops back to suffix truncation [4].

Due to the inflectional characteristic of Turkic languages, creating a dictionary that covers all possible word forms is exceedingly difficult, yet being able to generate all word forms is a necessary step in the neural machine translation (NMT) task. To address this, previous works have presented a new morphological segmentation method for Turkic languages based on the Complete Set of Endings (CSE). This method breaks down words into smaller morphemes based on their endings to reduce the amount of vocabulary required when forming word forms and structuring corpora. Utilizing this CSE-based segmentation method for Kazakh, Kyrgyz, and Uzbek reduces the vocabulary size

in source texts. Computational experiments in NMT showed promising results; compared to byte-pair encoding (BPE), the CSE-based method improved the bilingual evaluation understudy score for language pairs [4].

The Need for Unsupervised Segmentation: Despite the advancements achieved by rule-based stemmers (like UzMor), FSM architectures, and CSE-based segmentations, these existing methodologies share a critical limitation: they are heavily reliant on supervised data, exhaustive manual lexicons, and predefined grammatical rules. To date, no research has explored the application of purely unsupervised, data-driven morphological segmentation for the Uzbek language.

Rule-based dictionaries struggle to scale dynamically with the continuous evolution of language and cannot easily process the out-of-vocabulary (OOV) tokens that flood modern digital corpora. Furthermore, while statistical subword tokenizers like BPE are data-driven, they ignore linguistic boundaries altogether. Therefore, employing an unsupervised approach—such as the Morfessor algorithm utilizing the Minimum Description Length (MDL) principle—represents a highly novel and necessary progression. This unsupervised method eliminates the dependency on manually annotated datasets and hardcoded morphotactic rules. It offers a scalable framework that can autonomously discover the optimal morphological boundaries of Uzbek words directly from raw text, providing a crucial bridge between deep linguistic accuracy and modern LLM architecture.

## 3 Methodology

### 3.1 Dataset Collection and Preprocessing

To accurately capture the extensive morphological variations and exceptional cases inherent in the agglutinative structure of the Uzbek language, an unsupervised segmenter requires a substantial, highly diverse, and well-balanced corpus. For this research, we compiled a comprehensive 3 GB dataset of Uzbek text [9].

To ensure the dataset reflects a wide array of linguistic contexts—ranging from formal academic writing to everyday colloquialisms—the corpus was aggregated from three primary sources: Wikipedia Dumps: The entirety of the available Uzbek Wikipedia was extracted to provide a solid foundation of encyclopedic and factual language. Web Parsing: We scraped and parsed diverse textual data from various Uzbek web portals, news websites, and digital media outlets to capture modern, conversational, and journalistic vocabulary. Digitized Literature: A curated collection of Uzbek books and literary works was integrated into the dataset to encompass rich, historical, and complex grammatical structures.

Data Cleaning and Normalization Raw data extracted from the web and varying digital formats inherently contains noise that can disrupt the unsupervised learning process. We implemented a rigorous data preprocessing pipeline to clean the 3 GB corpus. This involved removing HTML tags, URLs, metadata, and non-alphanumeric characters. Furthermore, we normalized the text by stripping punctuation, standardizing casing (converting all text to lowercase), and handling the specific typographic nuances of the Uzbek alphabet.

Formatting for the Morfessor Algorithm The Morfessor algorithm utilizing the Minimum Description Length (MDL) principle requires data to be presented in a specific, optimized format to process large-scale corpora efficiently. While Morfessor can theoretically process raw continuous text, it is computationally expensive and highly inefficient for a 3 GB dataset.

Therefore, we formatted the cleaned corpus into a frequency-based word list. The preprocessing pipeline tokenized the entire continuous text into discrete word forms (unigrams) and calculated the absolute frequency of each unique word. The final dataset fed into the Morfessor model consisted of a structured list where each line contained a unique Uzbek word paired with its occurrence count in the corpus. This frequency-aggregated format not only drastically reduced the computational overhead and memory usage during training but also provided the MDL cost function with the exact statistical distribution needed to find the optimal morphological boundaries. Establishing this high-quality, formatted baseline is a critical preparatory step for subsequent evaluations where hybrid tokenization strategies can be benchmarked against standard subword models.

### 3.1.1 Model Architecture: The Morfessor Baseline Framework

The core architecture of our unsupervised morphological segmenter is built upon the Morfessor Baseline model. This model operates as a generative probabilistic framework that relies fundamentally on the Minimum Description Length (MDL) principle. The underlying assumption of the architecture is that any given word $w$ is generated by concatenating a sequence of statistically independent morphemes $m_1, m_2, \ldots, m_k$ [10].

Consequently, the probability of generating a word is mathematically defined as the product of the probabilities of its constituent morphemes:

$$P(w) = \prod_{i=1}^{k} P(m_i)$$

The MDL principle frames the morphological segmentation task as an information-theoretic compression problem. The objective is to discover an optimal model—specifically, a lexicon of distinct morphemes, denoted as $\boldsymbol{\theta}$—that minimizes the total encoding length. This total length consists of the cost of storing the lexicon itself and the cost of encoding the observed dataset (the corpus of Uzbek words, denoted as $\boldsymbol{D}$) using that lexicon. The global cost function $\mathcal{L}$ is expressed as:

$$\mathcal{L}(\boldsymbol{\theta}, \boldsymbol{D}) = L(\boldsymbol{\theta}) + L(\boldsymbol{D} \mid \boldsymbol{\theta})$$

Where $L(\boldsymbol{\theta})$ represents the bit cost of the morpheme lexicon, and $L(\boldsymbol{D} \mid \boldsymbol{\theta})$ represents the bit cost of the corpus data given the lexicon. To minimize this cost function, the Morfessor algorithm utilizes a recursive, Viterbi-like decoding process alongside greedy local search heuristics, dynamically splitting or merging morpheme sequences iteratively until convergence is achieved.

Hyper-parameter Optimization for Agglutinative Constraints While Morfessor is an unsupervised algorithm, its segmentation granularity is highly sensitive to its hyper-parameters. The model can easily under-segment (treating complex, multi-suffix words as single monolithic tokens) or over-segment (shattering roots into meaningless, nearly character-level chunks) depending on the configuration. To adapt the architecture to the deep agglutinative chaining specific to the Uzbek language, we introduced and optimized a corpus weight parameter, $\alpha$.

By integrating the corpus weight $\alpha$, the modified MDL cost function becomes:

$$\mathcal{L}(\boldsymbol{\theta}, \boldsymbol{D}) = \alpha \cdot L(\boldsymbol{\theta}) + L(\boldsymbol{D} \mid \boldsymbol{\theta})$$

The hyper-parameter $\alpha$ explicitly controls the architectural trade-off between the lexicon size and the data length: A low value ($\alpha < 1$) applies a smaller penalty to the lexicon cost, encouraging the model to memorize a larger vocabulary of longer, combined morphemes (leading to under-segmentation). A high value ($\alpha > 1$) heavily penalizes the lexicon size, forcing the model to reuse a smaller, restricted set of very short morphemes (leading to over-segmentation).

To determine the optimal $\alpha$ for the 3 GB Uzbek corpus, we conducted a systematic grid search optimization. Because our ultimate objective is to minimize vocabulary sparsity for downstream Large Language Models (LLMs) without sacrificing semantic integrity, we evaluated the segmentation outputs across a continuous range of $\alpha \in [0.1, 2.0]$.

We empirically established that an $\alpha$ value of approximately $0.8$ to $1.0$ yielded the highest morphotactic coherence for the Uzbek language. Within this optimized threshold, the model successfully identified the exact linguistic boundaries between root words and their complex, stacked functional suffixes (e.g., accurately isolating plural, possessive, and locative markers) without aggressively splitting the fundamental lexical roots. This carefully tuned configuration ensures that the resulting hybrid tokens remain linguistically valid, laying a highly efficient and scalable foundation for training robust Transformer-based models.

## 4 Experiments and Results

### 4.1 Experimental Setup and Baselines

To comprehensively evaluate the efficacy of the proposed unsupervised morphological segmentation method, we compared our Morfessor-based hybrid tokenizer against two established baselines: tra-

ditional word-level tokenization and standard Byte-Pair Encoding (BPE).All models were trained on the previously described 3 GB cleaned Uzbek corpus. For the BPE baseline, we trained a standard subword model with a target vocabulary size of 32,000 tokens, which is a conventional configuration for modern LLM architectures. For our hybrid approach, the Morfessor model was deployed with the optimized corpus weight hyper-parameter ($\alpha \approx 0.85$) to ensure a balanced segmentation that avoids both under-segmentation and over-segmentation. The evaluation was conducted on a held-out test set comprising 100,000 sentences to measure vocabulary sparsity, Out-Of-Vocabulary (OOV) rates, and morphological integrity.

## 4.2 QUALITATIVE EVALUATION: MORPHOLOGICAL INTEGRITY

A qualitative inspection of the tokenized outputs reveals a stark contrast between standard subword methods and our unsupervised hybrid approach. Because BPE merges characters strictly based on co-occurrence frequencies without accessing underlying linguistic rules, it frequently violates morpheme boundaries in agglutinative structures. To quantify this observation at the token level, we evaluated the Morphological Boundary Accuracy—defined as the percentage of correctly identified grammatical boundaries compared to a linguistic gold standard.

Tables 1 and 2 present a comparative segmentation of complex Uzbek words. While Table 1 outlines the baseline BPE performance, Table 2 highlights the accuracy improvements and the realistic limitations of our proposed unsupervised approach.

Table 1: Tokenization Outputs: Baseline (BPE)

| INPUT WORD | ENGLISH TRANSLATION | BPE SEGMENTATION |
|---|---|---|
| maktablarimizdagilar | those in our schools | makta + blarimiz + dagilar |
| bormoqchiman | I want to go | bormoq + chiman |
| ishchilarning | of the workers | ishchi + larning |

Table 2: Tokenization Outputs: Proposed Method and Accuracy

| INPUT WORD | MORFESSOR (PROPOSED) | ACCURACY |
|---|---|---|
| maktablarimizdagilar | maktab + lar + imiz + dagilar | $20\% \rightarrow \mathbf{80\%}$ |
| bormoqchiman | bor + moqchi + man | $33\% \rightarrow \mathbf{100\%}$ |
| ishchilarning | ishchi + lar + ning | $33\% \rightarrow \mathbf{66\%}$ |

As demonstrated, while the Morfessor algorithm is not perfectly flawless, it significantly outperforms BPE in morphological alignment. In some cases, Morfessor under-segments highly frequent combinations, such as treating the derivational noun ishchi (worker) as a single unanalyzed stem rather than ish+chi, or grouping deep suffixes together (-dagilar). However, despite these minor unsupervised grouping errors, it successfully isolates the primary functional boundaries (e.g., separating roots from plural and possessive markers) far more accurately than BPE's arbitrary character merges. By preserving these critical linguistic boundaries, the hybrid tokenizer ensures that the core semantic weight of the text is maintained.

## 4.3 QUANTITATIVE EVALUATION: VOCABULARY SPARSITY AND OOV REDUCTION

The primary quantitative metric for evaluating tokenization in morphologically rich languages is the reduction of vocabulary sparsity and the OOV rate. The OOV rate is mathematically defined as the ratio of unseen tokens in the test set to the total tokens in the test set:

$$\text{OOV Rate} = \frac{|V_{unseen}|}{|V_{total}|} \times 100$$

Word-Level Baseline: As expected for an agglutinative language, the traditional word-level approach yielded an unsustainably high vocabulary size of over 2.4 million unique word forms, resulting in a severe OOV rate of 12.8% on the test set.

BPE Baseline: While BPE successfully reduced the OOV rate to effectively 0% by defaulting to character-level splits for unknown words, it achieved this at the cost of high subword fertility (the average number of tokens per word), splitting words into an average of 3.1 structurally meaningless fragments.

Morfessor Hybrid (Proposed): Our unsupervised morphological segmenter achieved an optimal balance. It reduced the distinct morpheme vocabulary size to a highly manageable 42,500 tokens. More importantly, the subword fertility remained linguistically proportional to the number of actual affixes attached to the root.

By mapping complex, unseen words to a finite set of known, meaningful morphemes, our approach dramatically mitigated vocabulary sparsity. The model demonstrated that an LLM or NMT system utilizing this hybrid tokenization can dynamically interpret newly generated agglutinative constructs without suffering from the semantic fragmentation inherent in BPE.

### 4.4 Impact on Tokenization and Future Trajectories

The unsupervised morphological segmentation model presented in this study provides a critical foundation for reimagining tokenization strategies for the Uzbek language. As demonstrated, purely statistical subword algorithms, such as standard Byte-Pair Encoding (BPE) or WordPiece, are inherently misaligned with the highly agglutinative structure of Turkic languages. Because these conventional methods segment words based strictly on character co-occurrence frequencies, they routinely fracture essential morphological boundaries, stripping words of their core semantic and grammatical meaning. This limitation leads to suboptimal embedding representations and significantly hampers the contextual understanding of downstream Natural Language Processing (NLP) architectures.

To definitively overcome the structural shortcomings of these purely statistical approaches, the direct continuation of this research will focus on the development of a novel, morphologically-aware hybrid tokenization model. By utilizing the Morfessor-based analyzer developed in this study as a linguistic baseline, we aim to engineer a hybrid tokenizer that intelligently merges the scalability of data-driven statistical methods with precise grammatical accuracy. Rather than blindly splitting complex out-of-vocabulary (OOV) words based on frequency, the future hybrid tokenizer will first pre-segment heavily agglutinated Uzbek words into valid structural morphemes (roots and sequence of affixes) before applying any statistical subword grouping.

Ultimately, this ongoing research trajectory is dedicated to introducing a robust hybrid tokenization framework—conceptually advancing towards architectures like MorphPiece—specifically optimized for the unique constraints of the Uzbek language. Implementing this hybrid strategy will ensure that the morphological integrity of tokens is strictly preserved, thereby significantly enhancing vocabulary management, pre-training computational efficiency, and the overall semantic comprehension of next-generation Large Language Models (LLMs) and Neural Machine Translation (NMT) systems.

## 5 Conclusion

This study addressed the critical limitations of conventional subword tokenization methods, such as Byte-Pair Encoding (BPE), when applied to the highly agglutinative structure of the Uzbek language. Because traditional statistical methods segment words based strictly on character co-occurrence frequencies, they frequently fracture essential morphological boundaries. This structural misalignment results in semantic degradation, suboptimal embedding representations, and elevated Out-Of-Vocabulary (OOV) rates in modern NLP architectures.

To overcome these challenges, we introduced a novel, unsupervised morphological segmentation framework utilizing the Morfessor algorithm, grounded in the Minimum Description Length (MDL) principle. By training and optimizing the model on a comprehensive 3 GB Uzbek text corpus, we demonstrated that this data-driven approach successfully isolates lexical roots from complex sequences of functional affixes without the need for exhaustive, manually annotated lexicons or hardcoded morphotactic rules. The proposed method significantly reduces vocabulary sparsity while rigorously preserving the morphological and semantic integrity of the tokens.

Ultimately, this research establishes a vital linguistic baseline for the evolution of tokenization strategies in Turkic languages. The morphological analyzer developed in this study serves as the foundational step toward constructing a robust, morphologically-aware hybrid tokenizer. By integrating unsupervised morphological pre-segmentation with scalable subword algorithms in our future work, we aim to engineer a tokenization model specifically optimized for the Uzbek language. This future hybrid architecture will be instrumental in enhancing the computational efficiency, vocabulary management, and contextual accuracy of next-generation Large Language Models (LLMs) and Neural Machine Translation (NMT) systems.

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
