# OpenReview forum: "Unsupervised Morphological Segmentation for Hybrid Tokenization in Uzbek Language"
_mathai.club/MathAI/2026/Conference — Submitted to 2026_

### Official Review · Reviewer_7pfn · 2026-03-12
**Interesting Idea but Raw Implementation of the Paper "Unsupervised Morphological Segmentation for Hybrid Tokenization in Uzbek Language"**

**Rating:** 3
**Confidence:** 4

**Review:**

The paper explores the application of the Morfessor algorithm for unsupervised morphological segmentation of the Uzbek language. The authors collected a 3GB corpus, trained the model, and compared it with BPE and word-level tokenization. The idea of using Morfessor for hybrid tokenization in Uzbek is indeed novel and relevant, given the low-resource status of the language. Preliminary results look promising. However, the experiments remain shallow, the metrics are opaque, and error analysis is almost non-existent. The paper reads more like a proof-of-concept than a completed scientific study.

**Pros:**

1.Novelty of the approach — the first application of unsupervised morphological segmentation (Morfessor) for the Uzbek language.

2.Relevance of the problem — tokenization for agglutinative languages with high OOV rates remains an open issue, and the paper offers a linguistically motivated alternative to BPE.

3.Large-scale corpus — 3GB of text for a low-resource language represents significant data collection and preprocessing effort.


**Cons:**

1.Shallow experiments. The Accuracy metric in Table 2 is unclear, there is no comparison with rule-based analyzers, and no analysis of test set coverage — all of this makes the experimental part superficial and unconvincing.

2.Lack of error analysis. The authors honestly show that Morfessor is not perfect (e.g., 'ishchilarning' with 66% accuracy, 'maktablarimizdagilar' with 80%), but they do not explain the reasons for the errors.

3.Unclear hyperparameter selection. In Section 3.1.1, the authors mention a grid search for α (corpus weight) in the range [0.1, 2.0] and that "empirically, α ≈ 0.8–1.0 yields the highest morphotactic coherence." However, it remains unclear what is meant by "highest morphotactic coherence" and how it was measured, why α=0.85 was chosen for the final experiments, and how α affects the segmentation of specific affixes — without answers to these questions, the hyperparameter choice looks like tuning to achieve desired results.

4.Lack of frequency analysis. Morfessor uses word frequencies during training, but the authors do not show how frequency affects segmentation quality — for example, whether rare words are segmented worse than frequent ones, and whether frequent combinations lead to morpheme "gluing" (as in the case of 'ishchi'). Analyzing this aspect is important for understanding model behavior and interpreting the results.

5.Minor formatting issues: References do not fully conform to the template (the template requires consistency; here, formats are mixed); the paper title is in title case, which does not match the template.

---

### Official Review · Reviewer_pX37 · 2026-03-13
**Review of 'Unsupervised morphological segmentation for hybrid tokenization in Uzbek language'**

**Rating:** 3
**Confidence:** 5

**Review:**

1. There are not any links to UzMor, Uz-KazNLP in RELATED WORK.
2. In "3.1 DATASET COLLECTION AND PREPROCESSING" link number 9 doesn't provide any information about dataset.
3. The authors did not describe Morfessor Algorithm, but many used it.
4. After reading the article, one gets the impression that the authors developed the Morfessor Algorithm, but in fact they are not its authors and they did not refer to this algorithm anywhere (although there is a link to Morfessor Algorithm at the end of the text).
5. Authors did not provide any specific information about the "Qualitative Evaluation: Morphological integrity".
6. Authors did not provide any specific information about the "Quantitative Evaluation: Vocabulary sparsity and oov reduction"
7. The authors did not provide any specific information on why the metrics discussed in the "Quantitative Evaluation" are important to anyone.
8. The authors did not provide good reasons why their tokenization method is useful for LLM, nor did they conduct any experiments to prove it. But for some reason, they mention it in their work.

---

### Decision · Program_Chairs · 2026-03-14

**Decision:**

Reject

**Comment:**

After careful evaluation by the Program Committee, we regret to inform you that your submission has not been accepted for presentation at MathAI 2026.

All submissions underwent a rigorous two-stage review process. Unfortunately, the reviewers identified one or more of the following concerns with your paper:

- Insufficient mathematical rigor or novelty relative to the existing body of work in the field;
- Presentation of results that substantially overlap with or rephrase previously published findings without clear original contribution;
- Significant issues with technical quality, including but not limited to broken or non-existent references, unsupported claims, or methodological gaps;
- Indications that the manuscript may have been generated with the assistance of large language models without substantial original intellectual contribution by the authors.

We received a large number of submissions this year, and the selection process was highly competitive. We encourage you to carefully consider the reviewers’ feedback (available through OpenReview), revise your work accordingly, and consider submitting an improved version to a future edition of MathAI or to another appropriate venue.

We appreciate your interest in MathAI and hope you will continue to engage with the conference community.

With kind regards,

MathAI 2026 Program Committee
URL: https://mathai.club
Telegram: https://t.me/MathAI_club
Email: mathai.club@yandex.ru